# Analyzing the Association Between Depression and High-Risk Sexual Behavior Among Adult Latina Immigrant Farm Workers in Miami-Dade County

**DOI:** 10.3390/ijerph16071120

**Published:** 2019-03-28

**Authors:** Hyunjean Kim, Yingxin He, Ryan Pham, Gira J. Ravelo, Patria Rojas, Pura Rodriguez, Grettel Castro, Noël C. Barengo, Juan M. Acuña, Elena Cyrus

**Affiliations:** 1Department of Medical and Population Health Sciences Research, Herbert Wertheim College of Medicine, Florida International University, Miami, FL 33199, USA; yhe011@med.fiu.edu (Y.H.); rpham004@med.fiu.edu (R.P.); rodrigup@fiu.edu (P.R.); gcastro@fiu.edu (G.C.); nbarengo@fiu.edu (N.C.B.); jacuna@fiu.edu (J.M.A.); 2Center for Research on US Latino HIV/AIDS and Drug Abuse, Florida International University, Miami, FL 33199, USA; gravelo@fiu.edu; 3Department of Health Promotion and Disease Prevention, Robert Stempel College of Public Health and Social Work. Center for Research on US Latino HIV/AIDS and Drug Abuse, Florida International University, Miami, FL 33199, USA; proja003@fiu.edu; 4Department of Epidemiology, Robert Stempel College of Public Health and Social Work. Center for Research on US Latino HIV/AIDS and Drug Abuse, Florida International University, Miami, FL 33199, USA; ecyrusca@fiu.edu

**Keywords:** depression, sexual behaviors, risk, Latinas, AIDS, HIV

## Abstract

Latinas are often more affected by HIV due to their socio-economic and demographic profiles and are also less likely to receive proper mental health care. Latina immigrants are often even more vulnerable due to socio-economic and cultural factors that place them at higher risk. The current study seeks to examine the association between depression and risky sexual behaviors among adult Latina immigrants from a farm working community in South Miami-Dade County, (Florida, USA). Cross-sectional secondary data analysis was used for responses from a community-based participatory research (CBPR) study. Out of 234 Latina immigrants, 15% reported being depressed and 80% were reported as having engaged in risky sexual behavior. Although no association was found between depression and high-risk sexual behavior, significant secondary findings present associations between risky sexual behavior and low sexual relationship power, interpersonal violence, and relationship status. Implications for future research on depression and risky sexual behaviors among this population are discussed.

## 1. Introduction

The United States (US) Latinx population has reached a new high of 58 million, accounting for half of the nation’s growth. Yet, despite representing 18% of the US population [1], in 2016, Latinxs accounted for 26% of all new HIV diagnoses, making them disproportionately affected by HIV [2]. Moreover, in 2016, of the 7000 women who received an HIV diagnosis, 19% were Latinas, compared to 17% among their non-Latina White counterpart [2], placing Latinas at an even greater disadvantage for HIV infection.

Some of the factors that have placed Latina immigrants at such a disadvantage are language barriers, acculturation challenges, and economic pressures—factors associated with depression and risky sexual behavior [3]. These undesired mental health outcomes, like depression and risky sexual behavior, may be due to fear of deportation that deters them from accessing health care and preventative services—already a scarcity in many immigrant communities [4]. Even where proof of legal status is not required for service delivery, misinformation about rights to health care, lack of trust, and HIV stigma prevent this population from accessing much needed information and services that aid in the prevention of risky sexual behaviors and accessibility to mental health care [5].

Latina immigrants who experience mental health issues and risky sexual behavior are of particular concern because both factors have been demonstrated to have a cyclical relationship that also increases HIV infections [6,7,8]. A previous study also found that HIV-positive patients who received longer durations of antiretroviral therapy had a higher likelihood of risky sexual behaviors among HIV-positive patients [9]. Compared with other groups, Latinas are profoundly affected by HIV, and are receiving less treatment for mental health [5]. Some of the potential causes for HIV infection among Latinas include lack of HIV knowledge, low risk perception, and low self-efficacy [10]. Additionally, research has shown that depression may moderate [11], or even predict risky sexual behavior [12]. Most of the current literature focuses on adolescents [13] and men who have sex with men [14] and have often excluded adult Latinas [15]. Yet, the Latina population is particularly important to address since they face many unique cultural challenges including traditional gender norms like machismo and marianismo [5] that lend them to have poorer sexual health [10]. This imbalance is a result of the pervasive acceptance of machismo, that gives men more sexual freedom to have multiple sexual partners; and of marianismo, that women are not allowed to receive sexual education, much less, empowerment to request protection [5]. Despite the disparities and risks associated with depression and risky sexual behaviors among Latinas, to date, there is scant research that examines this population.

The current study aims to address these gaps in research by evaluating the association between depression and high-risk sexual behavior among adult Latinas living in a South Miami-Dade County farm working community.

## 2. Methods

### 2.1. Study Design and Participants

The present study is a cross-sectional secondary analysis using data collected from a community-based participatory research study funded by the National Institute on Minority Health and Health Disparities and the National Institute on Alcohol Abuse and Alcoholism. The study’s two aims were to (1) develop a protocol for HIV prevention among a community of Latina immigrants living in a farm working community, adapted from an intervention called Salud/health, Educación/education, Prevención/prevention, Autocuidado/self-care (SEPA), and to (2) empower and increase the Latina immigrant farmworker’s ability to respond to HIV infection at the local level [16].

SEPA is an evidence-based, culturally tailored HIV/AIDS behavior change intervention that targets sexually active heterosexual Latinas ages 18 to 44 [17]. The intervention is based on Social Cognitive Theory which explains that behaviors can be influenced by factors such as observation, self-efficacy, outcome expectancies, and interactions with the environment. SEPA uses demonstrations, role play, and other skill-building exercises in order to improve the self-efficacy of Latinas concerning condom use, condom negotiation, and partner communication.

Participants were recruited via flyers, local outreach, and referrals from participants. From March 2014 to January 2016, 234 participants were recruited from four local community agencies that provide services to Latina immigrants and the farm working community of South Miami-Dade. The three methods of recruitment were (1) active recruitment, where the on-site coordinator and outreach workers recruited in the communities, centers, and areas around the centers where participants usually met, such as churches, laundromats, grocery stores, etc. The on-site coordinator and outreach workers provided information to the study to potential participants and conducted the screening process if applicable. The second method was (2) passive recruitment, which used posted flyers that had the study contact phone number and contact person, and finally (3) word of mouth, where women who participated in the study referred friends or acquaintances about the study.

Eligibility requirements contained the following inclusion criteria: (i) female; (ii) identify as Hispanic or Latina; (iii) aged 18–50 years; (iv) sexually active within the past three months; (v) and lived in the US for 3–10 years. Exclusion criteria consisted of previous participation in a SEPA intervention or another structured HIV/STI prevention program in the past six months.

### 2.2. Variables

Depression was assessed through the Patient Health Questionnaire [18]*,* which is a three-page diagnostic instrument with 75% sensitivity and 90% specificity that can be used to screen for mental disorders corresponding to specific DSM-IV diagnoses. The Patient Health Questionnaire has a Cronbach’s α coefficient of 0.89, indicating good internal consistency [19]. Responses for 9 depressive symptoms were recorded based on whether participants were affected “nearly every day”, “more than half the days”, “several days”, or “not at all”.

High-risk sexual behavior was assessed through the Sexual History Scale, which classifies participants as high-risk based on several items: if the participant had sex with more than one partner in the past three months, had oral/vaginal/anal sex without using a condom, had intercourse with someone who they believe had multiple partners, used drugs or alcohol before or during intercourse, or exchanged money/drugs for sex [20]. Four items assessed frequency of vaginal and anal sex with primary sex partner or someone other than primary sex partner in the past three months (e.g., “How often did you have vaginal sex with your primary sex partner?”). Four items assessed frequency of condom use during sex in the past three months (e.g., “How often was a condom used for vaginal sex with your primary sex partner?”). Ratios between sex frequency and condom use frequency were calculated. Condom use was then dichotomously recoded as 1 = always used a condom in the past three months, 0 = did not always use a condom in the past three months.

Interpersonal violence (IPV) was assessed via, the Conflict Tactic Scale [21]. In regard to the Conflict Tactic Scale, the modified short-form version (CTS2) consisting of 39 items was used to score severity levels and mutuality types of partner violence. Analysis of Cronbach’s α coefficient for the CTS2 exhibited values above 0.70 f [22].

Demographic information on age, education level, relationship status, country of origin, current employment, and income were assessed via the SEPA Demographic Intake form. The age of participants ranged from 18–50 years old. Relationship status was defined as either in a relationship or not in a relationship. The education level of participants was measured as either less than high school or high school and beyond. Employment status was dichotomized into employed or not employed. The monthly income of participants was collected as a ratio variable of total monetary income in dollars per month but categorized into ordinal variables of above or below a set threshold of $2000.

Self-esteem was measured through the Rosenberg Self-Esteem Scale [23], which consisted of 10 items that are scored based on a method of combined ratings, and demonstrated good reliability and internal validity produces a continuous scale. Information about self-efficacy was obtained via the Self-efficacy scale [23]. The scale assessed individual competency in HIV/AIDS prevention and consisted of 7 items combined into a final score. Higher scores correlated with “strongly agree” responses among participants. HIV knowledge was measured through the HIV Knowledge Questionnaire [24] which utilized a 12-item true/false scale to denote areas of misconception regarding HIV risk competency. Sexual relationship power was analyzed based on the Sexual Relationship Power Scale [25], a 23-item tool used to assess for the two factors of Relationship Control and Decision-Making Dominance in a relationship dynamic. The scale was scored separately for each subscale, with Relationship Control on a 4-point Likert scale (1 = Strongly Agree, 2 = Agree, 3 = Disagree, and 4 = Strongly Disagree) and Decision-Making Dominance using a different method (1 = Your Partner, 2 = Both of You Equally, and 3 = You). Subscale scores were combined into three categories (“high”, “medium”, and “low”, levels of power) for the Sexual Relationship Power Scale. Self-silencing was measured through the Self Silencing Scale [26] which was composed of four subscales that assess for negative self-judgment, interpersonal behavior schema, and conceptions of depression. Comprised of 31 items, the scale scored from 31 to 155. Lastly, marianismo was measured via the Marianismo Beliefs Scale [27]. This instrument, which provides insight on the culture norms, beliefs, and interdependence of the sample population, employs 24 items to assess for five different subscales: family pillar, virtuous and chaste, subordinate to others, silencing self to maintain harmony, and spiritual pillar. Scoring was different for each subscale, with higher scoring indicating strong match between beliefs of each Latina and inherent values of marianismo.

### 2.3. Statistical Analysis

SPSS version 20 (IBM, Chicago, IL, USA) was used to analyze the data. A descriptive analysis portrayed the baseline characteristics of the studied population by depression status (Table 1). Percentages were used for categorical variables. Means and standard deviations were used for continuous variables with normal distribution while median and interquartile range were used for continuous variables with non-normal distribution. Bivariate analysis was used using chi-squared test to identify possible confounders by analyzing potential associations between baseline characteristics/confounders and depression as well as baseline characteristics/confounders and high-risk sexual behavior. The chi-square test described the sample based on exposure in the columns, and checked for imbalances among the columns. Collinearity diagnostic was performed to ensure independence of all independent variables. Finally, binary logistic regression was performed to compute unadjusted and adjusted odd ratios (OR and aOR) and corresponding 95% confidence intervals (CI) for the association between depression and high-risk sexual behavior. The statistical significance for both modeling and final outcomes was noted at an alpha level of ≤0.05.

### 2.4. Ethical Considerations

The dataset used did not violate HIPAA regulations and was approved by the Institutional Review Board of Florida International University (IRB-13-0187-AM12). Additionally, this study presented secondary analysis on de-identified data, which serves to protect the personal information of study participants.

## 3. Results

The total sample population consisted of 234 adult Latina immigrants living in a farm working community, and of these, 36 reported being depressed. The age for the participants ranged from 18 to 50 years old. Table 1 presents the distributions of characteristics of the population according to depression status. The mean age was similar for those who were depressed (33.5 years) and not depressed (34.2 years). Relationship status was differently distributed according to depression (*p* < 0.05). The proportion of Latinas not in a relationship was larger among those that reported depression. Out of 36 Latinas who reported depression, 47.2% had low sexual relationship power (SRP) and 14% had high SRP; out of 196 Latinas who did not report being depressed, 21.9% had low SRP (*p* < 0.01) and 38.3% had high SRP (*p* < 0.01). The distributions for Self-Esteem and Self-Silencing according to depression was found to be statistically significantly different (*p* < 0.001). On average, Latinas who did not report depression scored higher on the Self-Esteem construct (32.3) than those who reported being depressed (28.4), who also reported higher average scores on the Self-Silencing scale (93), compared to those without depression (75.4).

Table 2 illustrates the distribution of potential confounders according to risky sexual behavior. Risky sexual behavior included having multiple partners in the past three months, oral/vaginal/anal sexual activity without a condom, sex with someone who they believed had multiple partners, sex under the influence, or sexual activity in exchange for money or drugs. Out of the 36 Latinas who reported being depressed, 32 (89%) engaged in risky sexual behavior. Out of the 196 Latinas not reporting depression, 156 (80%) engage in risky sexual behavior. However, the distribution of depression according to risky sexual behavior was not statistically significant. SRP, HIV knowledge, and self-efficacy were found to be differently distributed according to risky sexual behavior (*p* < 0.05).

Among all the Latinas with low SRP, 93.3% engaged in risky sexual behavior, compared to 78.5% of those with medium SRP, and 73.8% of Latinas with high SRP (*p* < 0.05). The mean score for HIV knowledge among all the Latinas who engaged in risky sexual behavior was 6.34 out of the total 12 possible points, as compared to a mean score of 7.40 among the women who did not engage in risky sexual behavior (*p* < 0.05). For Latinas who engaged in risky sexual behavior, the median self-efficacy score was 22.0, as compared to a median score of 24.0 among the Latinas who did not engage in risky sexual behavior (*p* < 0.001).

Table 3 presents the unadjusted and adjusted associations between depression and risky sexual behavior. Those that reported depression, had 1.77 times higher odds of engaging in risky sexual behavior than Latinas not reporting depression after adjustment, although this finding was not statistically significant in this study sample. However, Latinas who reported being in a relationship were 2.66 times more likely to engage in risky sexual behavior than those who reported not being in a relationship after adjustment (OR 2.66; 95% CI 1.17–6.01). Participants also had a 4.10-fold increased odds of engaging in risky sexual behavior when they had low SRP as compared to high SRP after adjustment (OR 4.10; 95% CI 1.06–15.96). Latinas facing IPV were 2.70 times more likely to engage in risky sexual behavior than those who did not experience IPV (OR 2.70; 95% CI 1.02–5.09). After adjustment, for each additional question answered correctly in the HIV Knowledge Questionnaire, Latinas were 16% less likely to engage in risky sexual behaviors (OR 0.84; 95% CI 0.71–1.00).

## 4. Discussion

Our study found Latina recent immigrants from this farm working community were significantly more likely to engage in risky sexual behavior when they reported lower sexual relationship power (SRP); when they experienced interpersonal violence (IPV); and when they were in a relationship. One surprising finding is the lack of statistical significance found in the association between depression and risky sexual behavior, a relationship that the literature suggests in other populations, such as adolescents [12] and men who have sex with men [13]. This may be due to a lack of power due to a small sample size. The prevalence of depression among Latina immigrants is 15.5% and a meta-analysis calculated an aggregate prevalence of 15.6% among migrants worldwide [28]. 

Our findings are largely consistent with the literature, revealing that Latinas, in comparison with non-Hispanic White women and other minority groups, disproportionately engage in high-risk sexual behavior when they have low SRP and face IPV [5,10]. Furthermore, Latina immigrants are at increased vulnerability to exploitation due to their immigration status [4]. An important generalization that can be made from these findings is the cultural impact of marianismo and machismo. Since the Latinx gender norms (machismo) encourage Latinos to have multiple partners, despite their relationship status, this practice becomes risky for both the traditional Latino and his partner. Heterosexual contact from primary partners is the number one mode of HIV transmission for Latina women. Thus, understanding the socio-cultural factors and characteristics associated with this population should compel the medical community to promote mental health awareness and culturally appropriate interventions. Results show that interventions need to address the traditional gender norms, health disparities, and socio-cultural factors Latina immigrants face when developing HIV preventions strategies. Although our hypothesis did not show statistical significance, results indicate that it may be due to a lack of power.

The important secondary findings of the study were linked to the potential confounders of SRP, HIV Knowledge, and self-efficacy as distributions according to presence or absence of risky sexual behavior. It is possible that low SRP and low self-efficacy fundamentally contribute to an increased emotional fragility and fatalism, and inability to defend against unwanted sexual advances, which might lead to higher rates of risky sexual behavior. Additionally, individuals with low self-efficacy often are more susceptible to usage of recreational drugs and negative societal influences that may potentiate such behavior. In regard to HIV knowledge, populations that rate low on sexual education level, such as the immigrant Latina population in this study, may be more prone to impulsive sexual behavior because they are unaware of the negative consequences regarding contraction of STIs as well as other harmful health outcomes.

The current study had predictive limitations in its cross-sectional design; temporal association and causality for the significant findings cannot be determined. Since the sample is a hard to reach population, snowball sampling was used, which removes the benefit of random sampling, making the findings less generalizable and possibly resulting in a non-representative sample. All three forms of recruitment are forms of nonprobability sampling. Data for the survey was conducted through self-reported interviews, which is often subject to social desirability and recall errors. Furthermore, this study did not report other confounding factors including alcohol use of participants, as an isolated variable, or other viral infections which are known to be associated with risk behaviors [29]. Although there was an effect seen between risky sexual behavior and depression, due to small sample size, the current study may be underpowered. 

## 5. Conclusions

Our study can help pioneer the growth of additional literature for this particular topic at the research front, as prior search of key terms related to the study yielded few results. The significance of relationships analyzed in the study demonstrates social relevance among the local Miami-Dade community due to the prevalence of low-income Latina populations in the area. Recent statistics provided by the Centers for Disease Control and Prevention present that South Florida has the highest incidence of HIV transmission [30], with Miami-Dade County as number one in the state for STI rates [31]. These figures pose an extreme problem in the local community because at-risk Latinas face many healthcare as well as cultural barriers that preempt them to low self-efficacy and low SRP. Our study reinforces a need for increased availability of self-empowerment resources for this population not limited to creating bilingual initiatives such as sexual education workshops at community health centers, or training for social workers to be more cognizant of providing social support for at-risk Latinas. Furthermore, while there was no significant association between depression and high-risk sexual behavior among adult Latina immigrant farm workers in Miami-Dade County, the results highlighted a correlation between high scores on the HIV Knowledge Questionnaire and low Risky Sexual Behavior. This presents an opportunity to utilize the HIV Knowledge Questionnaire as a tool for health care service personnel to identify patients for sexual deviance and other risky sexual behaviors, and tailor preventative care involving higher usage of sexual education and support systems. In addition, IPV was associated with higher risky sexual behavior, therefore, this can present as an alternative or supplement to identifying at-risk Latina populations when other tools are unavailable. Ultimately, this research can compel the medical community to implement more effective interventions in order to decrease the prevalence of HIV among adult Latina recent immigrants. Future studies may study the association between depression and risky sexual behavior in a larger sample size.

## Figures and Tables

**Table 1 ijerph-16-01120-t001:** Characteristics of adult Latina immigrant farm workers by depression status in SEPA study in Miami-Dade County in South Florida in 2014–2016.

	Depressed (*n* = 36)	Not Depressed (*n* = 196)	*p*-Value
Characteristics	*n* (%)	*n* (%)	
Age in Years (mean, SD)	33.50 (8.50)	34.2 (7.80)	0.64
Relationship Status			0.04
In a Relationship	24 (66.70)	160 (81.60)	
Not in a Relationship	12 (33.30)	36 (18.40)	
Education			0.09
Less than High school	24 (66.70)	101 (51.50)	
High school and beyond	12 (33.30)	95 (48.50)	
Employment			0.16
No	13 (36.10)	96 (49.00)	
Yes	23 (63.90)	100 (51.00)	
Monthly Income			0.31
<$2000	20 (57.10)	128 (66.00)	
≥$2000	15 (42.90)	66 (34.00)	
Sexual Relationship Power			<0.01
Low	17 (47.20)	43 (21.90)	
Medium	14 (38.90)	78 (39.80)	
High	5 (13.90)	75 (38.30)	
IPV			0.16
Yes	28 (77.80)	129 (65.80)	
No	8 (22.20)	67 (34.20)	
	M (SD)	M (SD)	
HIV Knowledge	5.78 (2.74)	6.67 (2.55)	0.06
Self-Esteem	28.42 (4.81)	32.34 (4.57)	<0.01
Self-Silencing	93.03 (13.85)	75.37 (16.9)	<0.01
Marianismo	63.44 (9.24)	58.78 (9.57)	0.06
	Median (IQR)	Median (IQR)	
Self-Efficacy	21 (18.25–25)	22.5 (19–25)	0.31

*n* (%) = number(percentage); M (SD) = Mean (standard deviation); IQR = interquartile range; IPV = interpersonal violence.

**Table 2 ijerph-16-01120-t002:** Characteristics of adult Latina immigrant sample by risky sexual behavior in SEPA study in Miami-Dade County in South Florida in 2014–2016.

	No Risky Sexual Behavior (*n* = 45)	Risky Sexual Behavior (*n* = 188)	*p*-Value
	*n* (%)	*n* (%)	
Age (mean, SD)	34.1 (7.90)	34.2 (8.00)	0.96
Depression			0.19
Yes	4 (11.10)	32 (88.90)	
No	40 (20.40)	156 (79.60)	
Relationship Status			0.05
In a Relationship	31 (16.80)	154 (83.20)	
Not in a Relationship	14 (29.20)	34 (70.80)	
Education			0.27
Less than High school	21 (16.70)	105 (83.30)	
High school and beyond	24 (22.40)	83 (77.60)	
Employment			0.75
No	22 (20.20)	87 (79.80)	
Yes	23 (18.50)	101 (81.50)	
Monthly Income			0.09
<$2000	34 (22.80)	115 (77.20)	
≥$2000	11 (13.60)	70 (86.40)	
Sexual Relationship Power			0.01
Low	4 (6.70)	56 (93.30)	
Medium	20 (21.50)	73 (78.50)	
High	21 (26.20)	59 (73.80)	
IPV			0.11
Yes	26 (16.50)	132 (93.50)	
No	19 (25.30)	56 (74.70)	
	M (SD)	M (SD)	
HIV Knowledge	7.40 (2.10)	6.34 (2.70)	0.01
Self-Esteem	31.76 (4.40)	31.71 (4.90)	0.96
Self-Silencing	74.27 (15.70)	79.09 (18.00)	0.10
Marianismo	58.16 (9.60)	59.87 (9.70)	0.29
	Median (IQR)	Median (IQR)	
Self-Efficacy	24.00 (22.00–25.5)	22.00 (19.00–25.00)	<0.01

*n* (%) = number (percentage); M (SD) = Mean (standard deviation); IQR = interquartile range; IPV = interpersonal violence.

**Table 3 ijerph-16-01120-t003:** Unadjusted and adjusted associations between depression and risky sexual behavior in adult Latina immigrant farm workers in the SEPA study in Miami-Dade County in Florida in 2014–2016.

	OR ^1^ (95% CI ^2^)	aOR ^3^ (95% CI)
Depression		
No	Ref ^4^	
Yes	2.05 (0.69–6.14)	1.77 (0.51–6.16)
Age in Years	1.00	
Relationship Status		
Not in a relationship	Ref	
In a relationship	2.05 (0.98–4.25)	**2.66 (1.17–6.01)**
Education		
High school and beyond	Ref	
Less than high school	1.45 (0.75–2.78)	1.01 (0.46–2.18)
Employment		
Yes	Ref	
No	0.90 (0.47–1.73)	
Monthly Income		
≥$2000	Ref	
<$2000	0.53 (0.25–1.12)	
Sexual Relationship Power		
High	Ref	
Medium	1.30 (0.64–2.62)	1.20 (0.53–2.72)
Low	4.98 (1.61–15.43)	**4.10 (1.06–15.96)**
IPV ^5^		
No	Ref	
Yes	1.72 (0.88–3.36)	**2.70 (1.02–5.09)**
HIV Knowledge	0.84 (0.73–0.97)	0.84 (0.71–1.00)
Self-Esteem	1.00 (0.93–1.07)	
Self-Silencing	1.02 (1.00–1.04)	0.99 (0.96–1.02)
Marianismo	1.02 (0.99–1.05)	1.00 (0.96–1.04)
Self-Efficacy	0.84 (0.76–0.93)	

^1^ odds ratio; ^2^ confidence interval; ^3^ adjusted odds ratio; ^4^ reference group; ^5^ interpersonal violence.

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
