# Peer review of "Analyzing the Association Between Depression and High-Risk Sexual Behavior Among Adult Latina Immigrant Farm Workers in Miami-Dade County"

_ijerph, 2019, doi:10.3390/ijerph16071120_

Round 1

Reviewer 1 Report

Thank you for inviting me to review “Analyzing the Association between Depression and High-risk Sexual Behavior among Adult Latina Immigrant Farm Workers in Miami-Dade County”. I have the following recommendations:

1.     The authors stated that “Latina immigrants who experience mental health issues and risky sexual behavior are of particular concern because both factors have been demonstrated to have a cyclical relationship that also increases HIV infections [6-8]”. It is important to explain more cyclical relationship. Please add the following statement:

…. Includes HIV infections [6-8]. Previous study also found that HIV-positive patients who received longer durations of antiretroviral therapy had a higher likelihood of risky sexual behaviors among HIV-positive patients [Reference: PMID: 29844289]

2.     Under results, the authors stated “196 Latinas who did not report being dressed”. The word dressed is not correct. It should be depressed.

3.     Under Table 1, 2 and 3, please define IPV.

4.     Under discussion, the authors should comment whether the prevalence of depression of this study is comparable to global prevalence of depression in immigrants worldwide. Please add the following statement at the second paragraph under discussion:

In this study, the prevalence of depression among Latina immigrants is 15.5%. A meta-analysis calculated an aggregate prevalence of 15.6% among migrants worldwide (Please cite a reference from Pubmed).

5.     Under limitations, please state that there are other confounding factors which are not explored by this study. Please add the following statement:

…. social desirability and recall errors. Furthermore, this study did not report other confounding factors including alcohol use of participants and other viral infection which are known to be associated with risk behaviors (Reference: PMID: 30781486).

Author Response

We are sincerely grateful to the reviewers for taking the time to evaluate our manuscript. All the comments were extremely helpful in improving the quality of the paper. Below please note our corrections and responses that fully address all reviewer comments.

Comment 1:

The   authors stated that “Latina immigrants who experience mental health issues   and risky sexual behavior are of particular concern because both factors have   been demonstrated to have a cyclical relationship that also increases HIV   infections [6-8]”. It is important to explain more cyclical relationship.   Please add the following statement:

…. Includes HIV infections [6-8].   Previous study also found that HIV-positive patients who received longer   durations of antiretroviral therapy had a higher likelihood of risky sexual   behaviors among HIV-positive patients [Reference: PMID: 29844289]

Response 1:

We have now added the following   sentence and added the reference:

Previous study   also found that HIV-positive patients who received longer durations of antiretroviral   therapy had a higher likelihood of risky sexual behaviors among HIV-positive   patients [Reference: PMID: 29844289].

Comment 2:

Under   results, the authors stated “196 Latinas who did not report being dressed”.   The word dressed is not correct. It should be depressed.

Response 2:

Error has been corrected.

Comment 3:

Under   Table 1, 2 and 3, please define IPV.

Response 3:

We   have defined IPV in tables 1 and 2. Table 3 already had IPV defined.

Comment 4:

Under   discussion, the authors should comment whether the prevalence of depression   of this study is comparable to global prevalence of depression in immigrants   worldwide. Please add the following statement at the second paragraph under   discussion:

In this study, the prevalence of   depression among Latina immigrants is 15.5%. A meta-analysis calculated an   aggregate prevalence of 15.6% among migrants worldwide (Please cite a   reference from Pubmed).

Response 4:

As   suggested, statement has been added along with the following reference:

Foo, S. Q.; Tam, W. W.; Ho, C. S.;   Tran, B. X.; Nguyen, L. H.; McIntyre, R. S.; Ho, R. C. Prevalence of   Depression among Migrants: A Systematic Review and Meta-Analysis. International   journal of environmental research and public health 2018, 15,   1986.

Comment 5:

Under   limitations, please state that there are other confounding factors which are   not explored by this study. Please add the following statement:

…. social desirability and recall   errors. Furthermore, this study did not report other confounding factors   including alcohol use of participants and other viral infection which are   known to be associated with risk behaviors (Reference: PMID: 30781486).

Response 5:

As   suggested, the statement and reference has been added.

Reviewer 2 Report

Major concerns:

1) The title of the manuscript stated “adult Latina immigrant”. However, the enrollment criteria are only “female”, which does not represent the adult Latina population since the male are missing. Why? If the data only contain female, then the title of the manuscript need to specifically mention that it is only analyzing female. The discussion and conclusion also need to be changed to reflect this point.

2) The P-values in all tables are confusing as to which particular groups are being compared. For example, relationship status p= 0.04, is it referring to “in a relationship” between the depressed and not depressed? Or is it referring to “not in a relationship” between the depressed or not depressed? Same problem throughout the entire manuscript. Please clarify what you are comparing.

3) P-value for SRP is also confusing. Same issue as above, what are the authors comparing? There are 3 separate groups within the SRP, therefore 3 set of P-values should be presence. Each comparing to their respective counterpart in the depressed or not depressed.

4) Please reanalyze the HIV knowledge, self-esteem, self-silencing and Marianismo in median rather than mean (in both tables 1 and 2). Why the P-value for the mean self-esteem is <0.01 when the standard deviation between the groups clearly overlapped?

5) Why the sample size in Table 2 is 233 when the total sample size is only 232?

6) Sample size of depression under no risky sexual behavior is 44, but the sample size for that whole group is 45? Please reanalyze the data carefully.

7) Delete the sentence about PubMed search.

8) Although the topic of this manuscript is about analysis of depression and sexual behavior, surprisingly the discussion did not discuss anything about depression and sexual behavior at all. The relationship and findings between the two need to be discuss properly.

9) Need to state clearly in the conclusion that there is no significant association between depression and high-risk sexual behavior among the Latina population in that specific area, which is the main finding of this manuscript.

Minor concerns:

1) Use the term “Latina” throughout the paper.

Author Response

We are sincerely grateful to the reviewers for taking the time to evaluate our manuscript. All the comments were extremely helpful in improving the quality of the paper. Below please note our corrections and responses that fully address all reviewer comments. The attached document contains responses to comments from both Reviewer 1 and 2. Responses to Reviewer 2's comments can be found after responses to Reviewer's 1's comments.

Comment 1: 
  The title of the manuscript stated “adult Latina immigrant”. However, the   enrollment criteria are only “female”, which does not represent the adult   Latina population since the male are missing. Why? If the data only contain   female, then the title of the manuscript need to specifically mention that it   is only analyzing female. The discussion and conclusion also need to be   changed to reflect this point.

Response 1:

The   noun “Latina” refers to a girl or woman of Latin American origin or descent   that is commonly used in the North American region. It is the feminine version   of “Latino.”

Comment 2:
  The P-values in all tables are confusing as to which particular groups are   being compared. For example, relationship status p= 0.04, is it referring to   “in a relationship” between the depressed and not depressed? Or is it referring   to “not in a relationship” between the depressed or not depressed? Same   problem throughout the entire manuscript. Please clarify what you are   comparing.

Response 2:

The   Chi-Square statistic is used for testing relationship between variables. The   null hypothesis of the Chi-Square test is that no relationship exists on the   categorical variables in the population; they are independent. If the p-value   is less than 0.05, we can conclude that the variables are not independent of   each other and there is statistical relationship between the categorical   variables. The differences that you observed between the cells in the   crosstabulation are probably too great to be attributed to chance, so the   data support the hypothesis of a difference. In addition, the goal of this table   is to describe the sample based on the exposure (depression) and check for imbalance   of among the columns (the categories that defined Depression (Yes vs. No).

To   clarify, the element has been added in the Methods section 2.3 Statistical analysis:

The   chi-squared test described the sample based on exposure in the columns, and   checked for imbalances among the columns.

Comment 3:
  P-value for SRP is also confusing. Same issue as above, what are the authors   comparing? There are 3 separate groups within the SRP, therefore 3 set of   P-values should be presence. Each comparing to their respective counterpart   in the depressed or not depressed.

Response 3:

The   same as Response 2.

Comment 4:

Please   reanalyze the HIV knowledge, self-esteem, self-silencing and Marianismo in   median rather than mean (in both tables 1 and 2). Why is the P-value for the   mean self-esteem <0.01 when the standard deviation between the groups   clearly overlapped?

Response 4:

Medians   and IQR were reported for continuous variables with non-normal distribution.   We reported means and SD for HIV Knowledge, self-esteem, self-silencing and marianismo because they resembled a   normal curve.

In   regards to the p-value being <0.01 and the overlapping of SD, the presence   or absence of overlapping in the standard deviations, does not necessarily   correspond to a significant difference in the means being compared.

Comment 5:

Why   is the sample size 233 in Table 2 when the total sample size is only 232?

Response 5:

According   to the outputs, the total number of individuals eligible for the study is 234.   Table 1 has a total of 232 because 2 participants had missing information on   “depression”. Table 2 has a total of 233 because 1 participant had missing   information on “risky sexual behavior”

Comment 6:

Sample   size of depression under no risky sexual behavior is 44, but the sample size   for that whole group is 45? Please reanalyze the data carefully.

Response 6:

As   mentioned in Comment 5, out of the 45 individuals with “no risky sexual   behavior”, one had missing information on “depression”; therefore, the   discrepancy between the counts.

Comment 7:

Delete   the sentence about PubMed search.

Response 7:

As   suggested, sentence has been deleted.

Comment 8:

Although the topic of this manuscript is about analysis of depression   and sexual behavior, surprisingly the discussion did not discuss anything   about depression and sexual behavior at all. The relationship and findings   between the two need to be discuss properly.

Response 8:

This element has been addressed by the following:

One surprising finding is the lack of statistical significance found   in the association between depression and high-risk sexual behavior, a   relationship that the literature suggests in other populations such as   adolescents [12] and MSM [13]. This result may be due to a lack of power due   to a small sample size. The prevalence of depression among Latina immigrants   is 15.5%, and a meta-analysis calculated an aggregate prevalence of 15.6%   among migrants worldwide [28]. 

Comment 9:

Need to state clearly in the conclusion that there is no significant   association between depression and high-risk sexual behavior among the Latina   population in that specific area, which is the main finding of this   manuscript.

Response 9:

This element has been added to the conclusion:

Furthermore, while there is no significant association between   depression and high-risk sexual behavior among adult Latina immigrant farm   workers in Miami-Dade County, the results highlighted a correlation between   high scores on the HIV Knowledge Questionnaire and low Risky Sexual Behavior.

Comment 10:

Use the term “Latina” throughout the paper.

Response 10:

Thank you for the comment. The term “Latina” has been incorporated   throughout the paper where it is relevant. Additionally, please note that the   noun “Latinx” is referring to gender-neutral individuals of Latin American descent   and is a nonbinary alternative to Latino or Latina.

Round 2

Reviewer 2 Report

none